# Assessing the Potential Usefulness of FDG LAFOV-PET for Oncological Staging: An Evaluation of Lesion Number and Uptake

**DOI:** 10.3390/cancers17121927

**Published:** 2025-06-10

**Authors:** Valentino Dragonetti, Sara Peluso, Gastone Castellani, Stefano Fanti

**Affiliations:** 1Nuclear Medicine, Alma Mater Studiorum-University of Bologna, 40126 Bologna, Italy; s.fanti@unibo.it; 2Department of Medical and Surgical Sciences, Alma Mater Studiorum-University of Bologna, 40126 Bologna, Italy; sara.peluso5@unibo.it (S.P.); gastone.castellani@unibo.it (G.C.); 3IRCCS Azienda Ospedaliero-Universitaria di Bologna, 40138 Bologna, Italy; 4Nuclear Medicine, IRCCS Azienda Ospedaliero-Universitaria di Bologna, 40138 Bologna, Italy

**Keywords:** LAFOV-PET, [^18^F]F-FDG PET/CT, diagnostic sensitivity, tumour-to-background ratio

## Abstract

[^18^F]F-FDG PET/CT is essential in the diagnostic work-up of numerous oncological conditions. However, it may fail to detect localisations with a low glucose metabolism. It is an intuitive assumption that Long Axial Field-Of-View–Positron Emission Tomography/Computed Tomography (LAFOV-PET/CT) would arguably have a higher sensitivity than Short Axial Field-Of-View (SAFOV) PET/CT scanners. However, it is still uncertain which patient cohorts may benefit the most from this new technology. The aim of the present analysis is to pinpoint clinical scenarios in which the use of LAFOV-PET might meaningfully influence treatment decisions. We identified a range of scans with a lesion number between 1 and 2, as well as a maximum Standardised Uptake Value (SUVmax) between 2 and 5; out of the 862 investigations we retrospectively analysed, 4.5% fit into this interval.

## 1. Introduction

Positron Emission Tomography/Computed Tomography (PET/CT) with [^18^F]F-FDG (F-fluorodeoxyglucose), amidst its countless applications, is a key diagnostic tool in the management of a wide range of oncological conditions [1,2].

However, despite the high sensitivity and specificity of conventional or Short Axial Field-of-View (SAFOV) PET/CT scanners, there are intrinsic limitations related to their spatial resolution [3] and their ability to detect lesions with a low glucose metabolism [4], and, therefore, with a low uptake of the radiopharmaceutical. Indeed, lesions of a small size or that are characterised by a low glucose uptake may elude detection [5], especially in areas with a high physiological background [1]. This may lead to an underestimation of the disease extent, negatively influencing clinical decisions and, consequently, patient prognosis.

In recent years, the introduction of Large Axial Field-Of-View (LAFOV) PET/CT scanners has paved the road for new perspectives relating to improving diagnostic performance [6]. These devices, with significantly greater axial coverage than SAFOV scanners, allow images of the entire body to be acquired in a single pass, arguably increasing the efficiency and sensitivity of the examination [7]. Indeed, the greater extension of the Field Of View (FOV) makes it possible to collect a greater number of annihilation events [8], improving the signal-to-noise ratio and potentially increasing the ability to detect lesions with a low uptake or small size [9].

Specifically, the distinction between SAFOV and LAFOV scanners is based on the extension of the axial FOV and the ability to cover larger portions of the body in a single acquisition. SAFOV scanners, with an extension between approximately 15 and 35 cm, require several bed positions to cover the distance between the cranial vertex and the thighs [10], which represents the body region that is most commonly acquired in PET/CT scans [11]. LAFOV scanners, which are characterised by an FOV greater than 100 cm, allow the interval from the vertex to the proximal femoral region to be acquired in a single bed position in most patients [10]. When the FOV extension exceeds 188 cm, as in the case of the uExplorer scanner (194 cm) [12], it is possible to obtain a total-body image, i.e., from the apex of the skull down to the toes, in a single acquisition in the majority of the population. This configuration is known as Total Body PET [10].

LAFOV-PET technology also offers the option to reduce the dose administered to the patient or to shorten the acquisition time [13], while maintaining an image quality that is comparable to or even better than that obtained with SAFOV scanners [7]. However, the aspect we rather intend to focus on is the potential increase in diagnostic sensitivity. An increase in sensitivity could, indeed, result in a more accurate disease characterisation, possibly influencing treatment choices and the consequent clinical outcomes.

In the framework of optimising healthcare resources, the adoption of new technologies, such as LAFOV-PET, must be justified by a concrete clinical benefit, taking into account the associated costs and resource availability. Therefore, it becomes crucial to identify the precise clinical settings in which the increased sensitivity offered by LAFOV-PET could provide a meaningful lead over SAFOV-PET.

Radiopharmaceutical uptake, expressed as maximum Standardised Uptake Value (SUVmax), is another key factor in lesion detection. Lesions with a low SUVmax are more challenging to identify and can be easily missed during acquisition with SAFOV scanners [4]. LAFOV-PET, due to its alleged higher sensitivity, could improve the detection of these low-uptake lesions, but it is necessary to understand in which clinical contexts this improvement could matter.

The current literature offers promising data on the potential of LAFOV-PET [14], but there is a lack of extensive studies clearly delineating the patient cohorts that could benefit most from this new technology.

In certain malignancies, the detection of occult metastases may dramatically modify the therapeutic approach [15]. Similarly, in patients with early-stage or oligometastatic disease, i.e., with a limited number of lesions, the detection of further localisations may have a greater influence [16]. Conversely, in patients with an extensively metastatic disease, the detection of additional lesions would not have such an impact on affecting the already-established treatment plan [17]. Indeed, according to lung cancer guidelines, additional nodules within the same lobe may still be approached surgically if technically feasible [18,19]. Conversely, discovering lesions in a different lobe or contralateral lung would shift therapy from surgical resection to systemic or multimodal treatments [18,19].

Therefore, it is vital to deepen our understanding of the clinical efficacy of LAFOV-PET, moving beyond mere technical considerations. Whether the increased diagnostic sensitivity has the potential to yield significant changes in therapeutic management needs to be assessed.

The aim of the present analysis is, therefore, to explore in which clinical settings the use of LAFOV-PET could provide a significant improvement in diagnostic accuracy relevant to patient management.

## 2. Materials and Methods

The present study was carried out retrospectively by analysing data collected between January 2024 and April 2024 at the PET centre of Ospedale Sant’Orsola in Bologna, Italy.

The key purpose was to assess in what proportion of patients suffering from oncological pathologies the new LAFOV-PET technology, due to its superior diagnostic sensitivity [6], could have a significant impact on the management of the therapeutic course of such patients compared to conventional SAFOV-PET scanners.

The study included patients undergoing [^18^F]F-FDG PET/CT for the first time who had come to our centre in a staging context. By staging context, we refer to both findings that had never been histologically characterised and patients who had just received a histological diagnosis but had not yet started any therapy. In particular, the main neoplasms included were head and neck cancer; cholangiocarcinoma (CCA); hepatocarcinoma (HCC); sarcomas; cancers of the colon, skin including melanomas, anus, oesophagus, breast, thyroid, ovary, pancreas, lung, kidney, rectum, stomach, adrenal, testis, and thymus; and haematological diseases such as multiple myeloma (MM) and lymphomas, both Hodgkin’s (HL) and non-Hodgkin’s (NHL), as well as either B-Cell Lymphoma (BCL) or T-Cell Lymphoma (TCL). The clinical questions relating to the investigation included staging, the clarification of doubtful findings revealed by other diagnostic methods, increased plasma tumour markers, or clinical suspicion of disease recurrence.

On the other hand, patients who had undergone PET/CT for restaging after therapy or for monitoring during therapy were excluded, as were patients who had undergone previous PET scans for oncological diseases. The rationale for not including this cohort of patients was that the primary diagnosis could not be ascertained and that the semi-quantitative values of SUVmax may have been influenced by prior therapies. Reports were also excluded if they lacked essential data, such as the SUVmax value of the lesion with maximum uptake.

The evaluators were an expert nuclear physician and a final-year nuclear medicine resident, who worked in parallel. All data were obtained through a manual review of PET/CT reports. They did not, therefore, review the images of the PET scans.

For each report, the number of pathological lesions identified and the SUVmax value of the lesion with the highest uptake were collected. In cases where the number of lesions exceeded the threshold of 20, the count was limited to this maximum value, given the minimal impact LAFOV-PET could have on these patients. The associated oncological pathology was also recorded, if available.

All PET scans whose report was analysed were performed on SAFOV scanners. The SAFOV scanners used for the present study are of three different types—GE Healthcare Discovery STE, GE Healthcare Discovery MI, and United Imaging Vista. Their FOVs are 15.7 cm [2], 25 cm [20], and 24 cm [21], respectively. The contribution of the different SAFOV scanners used was not evaluated in the current analysis. All scans were performed in compliance with EANM guidelines [22]. Patients fasted for at least 6 h before [^18^F]F-FDG injection (2–3 MBq/kg; uptake time: 60 ± 10 min). Low-dose CT (120 kV; 80 mA) was used for attenuation correction. On GE scanners (Discovery STE, Discovery MI), PET images were reconstructed with a 3D OSEM algorithm (2 iterations; 20 subsets), a 6 mm Gaussian filter, time of flight (TOF), and point spread function (PSF). For the Vista scanner, four bed positions (1 min/bed) were acquired, from head to feet; reconstructions included OSEM (2 iterations; 20 subsets; 3 mm smoothing) or Hyper DPR, both with TOF and PSF.

The proportion of patients in whom a LAFOV-PET could have changed the course of treatment was defined as those patients who had between 1 and 2 total lesions and an SUVmax value of the lesion with the highest uptake between 2 and 5. The rationale behind this choice was that in patients with numerous localisations, the finding of an additional site of disease would not have changed the treatment course. Furthermore, most scanners can identify lesions with high SUVmax, while a higher sensitivity might help in detecting those with a low metabolic uptake.

Data were collected via a text tabular data file, before being processed with the pandas and openpyxl Python v3.12 packages for data analysis. First, exploratory analyses (summary statistics) were conducted to retrieve statistical information about the variables of interest, including patient and lesion statistics. Then, a subset of scans was identified on the basis of lesion number and the SUVmax of the lesion with the highest uptake. This subset was then analysed in terms of statistical properties and in terms of the specific malignancy affecting the patients. Finally, plots were obtained using the Matplotlib v3.9 and Seaborn Python v0.13 packages.

We conducted the study in accordance with the Declaration of Helsinki and current data protection regulations. In accordance with the IRCSS (Istituto di Ricovero e Cura a Carattere Scientifico) regulations, the reports analysed by the two medical examiners did not contain any sensitive patient data, but merely the body of the report itself. Therefore, no patient could be identified.

## 3. Results

The primary criterion used to categorise each scan was whether it was negative or positive. A scan was classified as positive if both the number of lesions and the SUVmax were greater than 1.9, as well as the lowest value of SUVmax being reported as pathological. Among the 862 examinations, 574 (66%) fulfilled these criteria and were considered positive, while the remaining 288 (34%) were classified as negative and were excluded from further analysis. Upon analysis of the positive scans, 153 were found to feature exactly one lesion.

In total, 288 (33%) of the 862 scans were considered negative. Focusing on the positive scans, the number of lesions ranged from 1 upwards, with a mean of 7 ± 9 (standard deviation) and a median of 4. Out of the 574 positive scans, 55 presented 20 or more lesions. The SUVmax of the lesion with the highest uptake varied from 1.9 to 103.0, with a mean of 14 ± 10 and a median of 12. Among the positive scans, 32 had a highest-lesion SUVmax of 30 or more.

The distribution of the different conditions within the sample of 862 PET scans was investigated. The most common diseases included in the study included breast cancer (136; 15.8%), masses of undetermined nature (MUNs) (101; 11.7%), lung cancer (75; 8.7%), suspected lymphoma (51; 5.9%), head and neck cancer (48; 5.6%), multiple myeloma (42; 4.9%), and colon cancer (41; 4.8%). Other conditions, such as melanoma (32; 3.7%), sarcoma (19; 2.2%), uterine tumours (19; 2.2%), and bladder tumours (18; 2.1%) were less frequent, but still notable in the examined population.

Considering the most common diseases, the mean number of lesions and the mean SUVmax of the lesion with the highest uptake were (8, 12) for breast cancer, (7, 12) for MUNs, (7, 17) for lung cancer, (14, 15) for suspected lymphoma, (5, 15) for head and neck cancer, (12, 9) for multiple myeloma, (5, 13) for colon cancer, and (9, 18) for melanoma, respectively.

By considering cases with a number of lesions below 20 and an SUVmax of the lesion with the highest uptake of 30 or less, a subset of scans was selected to better visualise the distribution of these variables. Within this subset, a bubble plot was generated by discretising both lesion count and SUVmax. For the lesion count, lower values (e.g., 1, 2, and 3 lesions) were represented individually, while higher values were grouped into broader categories. The SUVmax was binned using a fixed increment of 3 units throughout the considered range (between 1 and 30, including extremes). The size of each data point in this enhanced scatter plot was proportional to the number of scans falling into the respective category, thus improving the overall readability. Moreover, the information about the number of scans belonging to each category is colour-coded. This bubble plot is shown in Figure 1.

A separate representation of the chosen ranges for SUVmax and lesion number is provided in Figure 2 and Figure 3, respectively. In these figures, the distributions are illustrated through count plots, and SUVmax values were rounded to facilitate discretisation along the *x*-axis.

Finally, the analysis focused on the aforementioned subgroup of scans where the higher sensitivity offered by LAFOV-PET might be more useful, previously defined as a lesion number in the range [1, 2] and an SUVmax of the lesion with the highest uptake in the range [2, 5]. This subset accounted for 4.5% of the entire set of 862 scans and 7% of the 574 positive scans. Among the most common malignancies observed in the study population, melanoma (2 out of 32; 6.2%), breast cancer (8 out of 136; 5.9%), and multiple myeloma (2 out of 42; 4.8%) showed higher-than-average proportions of cases meeting these criteria. Conversely, MUNs (4 out of 101; 4.0%), lung cancer (2 out of 75; 2.7%), head–neck cancer (1 out of 48; 2.1%), suspected lymphoma (1 out of 51; 2.0%), and colon cancer (0 out of 41; 0.0%) presented lower proportions. The percentages for each malignancy are referred to the total number of scans that were labelled with that specific malignancy. The corresponding bar plot is shown in Figure 4.

## 4. Discussion

The aim of the present study was to identify, in a preliminary way, those specific clinical contexts in which the increased diagnostic sensitivity offered by a LAFOV-PET scan could translate into a real therapeutic benefit. In particular, we focused on a range defined by the presence of 1–2 lesions and an SUVmax of the lesion with the highest uptake between 2 and 5. This choice reflected the hypothesis that in this narrow range, the increase in lesion detection capacity, and thus disease characterisation, could have the highest influence on patient management.

Although we focused on newly diagnosed patients in a pure staging context, we acknowledge that LAFOV scanners may also play a beneficial role in restaging or therapy monitoring settings. Its alleged increased sensitivity could, indeed, help detect minimal residual disease or early recurrence, potentially guiding therapeutic adjustments. About a third of all scans were negative, and LAFOV technology could arguably have a considerable impact on at least part of this cohort. Further studies would be required to quantify these advantages.

The reason behind the decision to use data obtained from SAFOV-PET to anticipate the possible impact of LAFOV-PET was dictated by a practical and rational methodological consideration. Indeed, we considered the current clinical scenario as a baseline and, based on these data, tried to outline an area in which LAFOV-PET technology could offer tangible added value.

The preference to limit the analysis to those patients with a number of lesions between 1 and 2 and an SUVmax of the lesion with the highest uptake between 2 and 5 was guided by both technical and clinical considerations. On the one hand, numerous lesions tend to suggest an already extensively metastatic setting, in which the identification of additional foci would not significantly alter the therapeutic strategy. On the other hand, lesions with higher uptakes (>5) are usually already easily detected by SAFOV scanners, making the hypothetical increased sensitivity of LAFOV-PET less relevant. On the contrary, the SUVmax range of 2–5 represents a diagnostic grey area, in which lesions present an uptake that is neither low enough to be completely invisible nor high enough to clearly emerge from the physiological background. Precisely in this range, an increase in sensitivity and a better Tumour-to-Background Ratio (TBR) could allow incipient or inactive metastases to be recognised, thus altering the patient’s prognostic and therapeutic framework.

Although established in clinical practice, SAFOV-PET scanners present physiological and technical limitations that reduce their sensitivity [23]. The limited axial FOV forces acquisition by body segments, with a lower event count and potentially worse signal-to-noise ratio, thus adversely affecting the detection of small lesions or lesions with a low glucose uptake [24].

Conversely, modern LAFOV-PET devices offer a significantly greater axial extension, allowing images of the entire body to be acquired simultaneously, thereby increasing the efficiency of photon capture, which could translate into increased sensitivity and the ability to discriminate minor lesions or lesions with a low uptake [6]. The emerging literature already describes concrete improvements obtained with these systems, including a reduction in acquisition time, the possibility of decreasing the dose administered to the patient while maintaining or even increasing image quality, and, above all, a greater ability to detect lesions with a lower TBR [25].

Evidence from our analysis suggests that such a sensitivity boost could translate into a definite benefit for a non-negligible fraction of patients. Even if we limit ourselves to the most stringent conditions identified (1–2 lesions; SUVmax 2–5), 4.5% of the entire population analysed, corresponding to 7% of positive patients, could benefit from a more sensitive investigation. In concrete terms, identifying an additional small lesion in borderline cases could shift the therapeutic approach from purely locoregional treatments to systemic therapy, if the disease is upstaged. For instance, in early-stage breast cancer or melanoma, detecting even a single low-uptake metastasis can prompt immediate modification of the surgical plan or an earlier introduction of systemic therapies [26,27]. Although this share is not large in absolute terms, the clinical relevance should not be underestimated, since each of these patients could see a substantial change in therapeutic management.

The presented analysis suffers from some inherent limitations. First, it is a retrospective study, based on data from patients undergoing SAFOV-PET, without actual direct comparison with LAFOV-PET being performed in parallel. In addition, the choice of criteria to define the subgroup of patients potentially benefitting from the highest level of sensitivity (1–2 lesions; SUVmax 2–5) derives from reasonable considerations, but remains, to some extent, arbitrary and to be verified in prospective settings and correlated with clinical and survival outcomes. Lastly, the study population, which is large in number but not extensive, reflects the case history of a single PET centre and may not be fully representative of other clinical settings. Furthermore, the use of a LAFOV-PET scan in the present analysis would likely have increased the proportion of total patients falling into the diagnostic area of uncertainty for SAFOV scanners; consequently, our presented results might be underestimated. Moreover, comparing data solely from historical SAFOV scanners prevents us from isolating the effect of LAFOV technology from other confounding factors such as patient heterogeneity or minor variations in imaging protocols. Different scanner generations, reconstruction algorithms, or patient selection biases could also have influenced lesion detectability. Therefore, while we propose that increased sensitivity from LAFOV scanners might benefit a subset of patients, our retrospective approach cannot exclude that other non-technical aspects contributed to the observed diagnostic gaps. Furthermore, in the current analysis, we have not evaluated each SAFOV scanner contribution and lesion detectability. Lastly, gathering data by reading the reports of numerous nuclear medicine physicians, although allowing for a more representative case history than that of the entire PET centre, brings with it a degree of heterogeneity in the classification of a lesion as positive or otherwise.

Let us now explore the potential role of LAFOV-PET in the most common oncological diseases in the examined population. Specifically, the conditions in which we noted that the contribution of LAFOV-PET might be higher than average (i.e., as previously defined, lesion number between 1 and 2, as well as an SUVmax between 2 and 5) are melanoma and breast cancer. The characterisation of multiple myeloma and masses of undetermined nature (MUNs) would be fundamentally in line with the general study population, whereas the contribution would be lower in lung, head and neck cancer, and lymphomas.

An analysis of the most common pathologies shows that the potential usefulness of the increased diagnostic sensitivity offered by LAFOV-PET is not uniform across clinical settings. In some neoplasms, even the identification of a single metastasis with moderate uptake can significantly change the therapeutic approach, while, in others, the impact is less pronounced, often because the disease already presents with a high burden of lesions or with uptake levels that make it detectable even with conventional scanners.

The instance of melanoma is rather surprising, since this pathology presented itself in the study population with a high average of both number of lesions and uptake [28]. However, there may be cases in which lymph node or metastatic localisations are of a size at the limit of the resolving power of classic SAFOV scanners [29]. Their identification would change the therapeutic strategy from locoregional surgery [30] to the early introduction of immunotherapies [31] or target therapies [32].

In breast cancer, the presence of metastases with a moderate SUVmax is not uncommon [33]. The discovery of a metastatic lesion, even a single one, may lead to a more radical mastectomy [34] instead of conservative surgery [35], or to the early integration of radiotherapy [36] or more aggressive systemic therapies, such as chemotherapy [37], targeted therapies, or immunotherapy [38], modifying the timing and intensity of the therapeutic approach.

Subgroups in which we did not find a significant difference to the general population are multiple myeloma and masses of undetermined nature (MUNs). However, even within the average of the general population, there is still almost one in twenty patients who could benefit from a greater sensitivity. MUNs, in particular, represent a heterogeneous set of diseases and this potentially explains why the proportion of patients in the previously identified ranges who came in for MUN characterisation was then similar to that of the general study population.

Finally, we addressed how in some clinical settings, such as lung cancer, head–neck cancer, colon cancer, and lymphomas, the proportion of patients falling into the previously defined diagnostic grey area was lower. In this subset of malignancies, lung, head–neck, and colon cancers had an average number of localisations that was modest, ranging from 5 to 7, whereas lymphomas in staging tended to have an already high number of localisations. What rather unites this heterogeneous group of pathologies is the mean SUVmax of the lesion with the highest uptake, which in none of these cases falls below a value of 13.

It is also noteworthy that the category “other malignancies”, which excludes the more common malignant diseases discussed in the previous sections, saw a proportion of patients in the area of diagnostic uncertainty at 5.7%, compared with 4.5% of the total population. This is probably due to the influence of common diseases such as lymphomas and lung cancer, which lowers the proportion of the population as a whole. We have also included malignancies that are typically characterised by lower or heterogeneous [^18^F]F-FDG uptake, such as HCC. The reason behind this was that these are settings where SAFOV scanners often underperform [4] and LAFOV scanners might offer added diagnostic value. However, these low-uptake neoplasms represented only a small minority of our overall cohort.

While these preliminary data suggest that nearly one in twenty patients could potentially benefit from improved sensitivity, they require confirmation in prospective, controlled studies. It will be crucial to conduct direct comparisons between SAFOV-PET and LAFOV-PET on the same patient and at the same time in order to validate the true diagnostic impact, assess the repeatability of the results, and consolidate the generalisability of the data. In regard to this fundamental new field of research, our operating unit is currently working on a study that aims to directly compare LAVOF and SAFOV scanner performance in multiple diagnostic settings.

Furthermore, larger samples will be needed, and the systematic collection of clinical endpoints such as survival, disease control, and quality of life will be required to quantify the true long-term clinical benefit. In parallel, the economic and organisational aspects of the large-scale introduction of LAFOV-PETs will also need to be considered in order to assess their sustainability and true cost–benefit ratio in routine clinical practice.

## 5. Conclusions

Our analysis indicates that 4.5% of the patients studied with [^18^F]F-FDG PET/CT for oncological conditions in a staging setting on SAFOV-type scanners presented a lesion number between 1 and 2 and a SUVmax between 2 and 5, and may have benefitted from the use of more-advanced LAFOV-PET scanners.

However, these results should be interpreted with caution, considering the aforementioned limitations of the study and the current absence of direct comparisons with LAFOV-PET. Therefore, it appears essential to set up larger prospective studies and to integrate clinical outcome data in order to confirm the real added value of these new technologies.

## Figures and Tables

**Figure 1 cancers-17-01927-f001:**
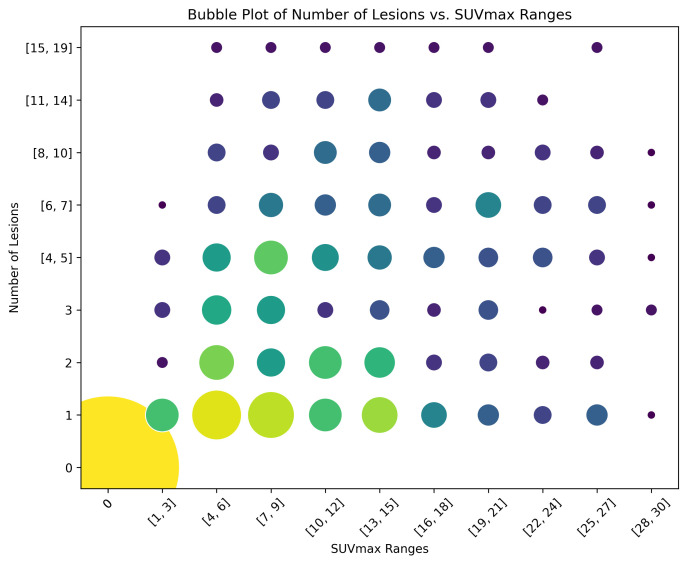
Bubble plot for the selected subset of scans. Each data point is both size-coded and colour-coded with respect to the number of scans falling within that category.

**Figure 2 cancers-17-01927-f002:**
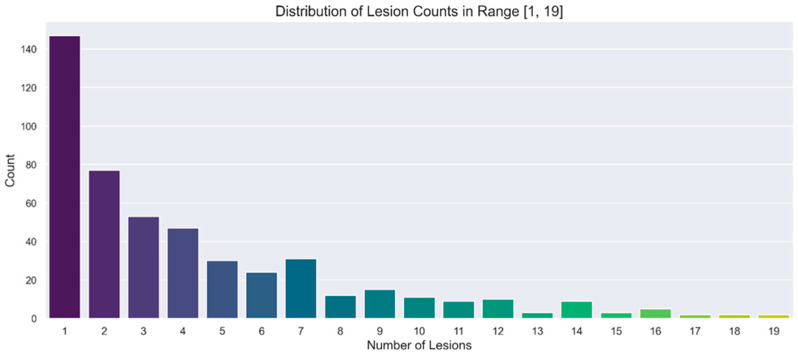
Count plot showing the distribution of number of lesions in the range of interest [1, 19].

**Figure 3 cancers-17-01927-f003:**
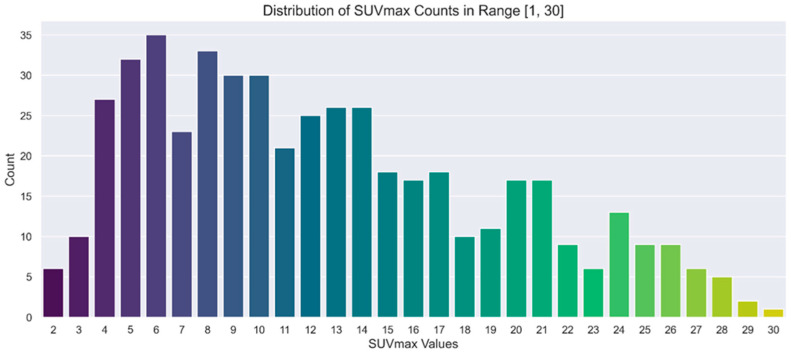
Count plot showing the distribution of SUVmax in the range of interest [1, 30].

**Figure 4 cancers-17-01927-f004:**
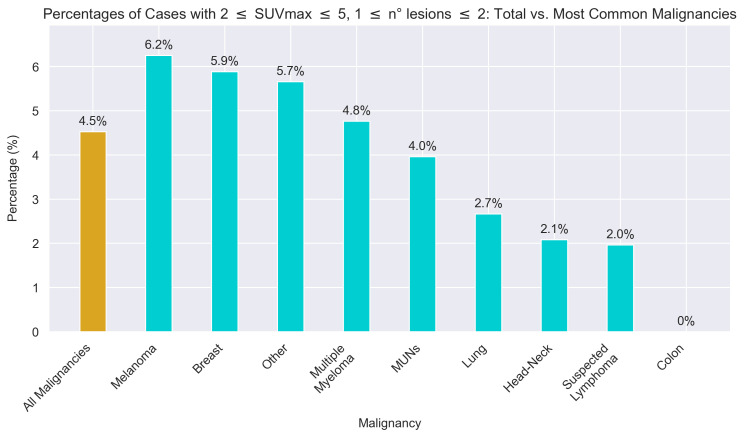
Bar plot showing the percentage of cases meeting the conditions relating to number of lesions and SUVmax of the lesion with the highest uptake for the most common malignancies examined.

## Data Availability

The original contributions presented in this study are included in the article. Further inquiries can be directed to the corresponding author.

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
