# Peer review of "Assessing the Potential Usefulness of FDG LAFOV-PET for Oncological Staging: An Evaluation of Lesion Number and Uptake"

_cancers, 2025, doi:10.3390/cancers17121927_

Round 1
Reviewer 1 Report (Previous Reviewer 1)
Comments and Suggestions for Authors
This version of the paper has been improved and all my comments have been addressed.
Author Response
I would take this opportunity to thank the reviewer, for their valuable contributions in improving the manuscript.
Reviewer 2 Report (New Reviewer)
Comments and Suggestions for Authors
This manuscript provides an interesting point of view for assessment of potential usefulness of LAFOV-PETs in a staging setting of multiple malignancies.
I must admit that I´m wondering why the authors did not collect the data of used scanner type. Very probably there is some identifier (e.g. the accession number for image) in the PET/CT report which could be easily related to the used scanner. And it would be only minor work to collect this data and assess the probable association between the scanner type and sensitivity performance. This assessment would be of great value strengthening the conclusions of the study.
However, the study data is straightforward and clearly presented in the manuscript. I have only a few minor comments:
- Line 153: A reference could be provided for the EANM guidelines (EARL accreditation?)
- The abbreviation SAFOV has now two meanings (please see Simple Summary and introduction section line 52–53).
- Lines 181–182: “A scan was classified as positive if both the number of lesions and the SUVmax value were greater than zero.” This was a retrospective study and only reports were evaluated. Thus, I suppose that a lesion with an SUVmax value e.g. lower than one, would not have been mentioned in the report and consequently would not have been classified as positive. If I am right, then please edit this sentence somehow.
- The study results are applicable for patients undergoing initial staging of cancer disease. It is stated several times in the manuscript but cannot be said too clearly or too many times (considering other indications for oncological PET/CT imaging, such as monitoring treatment response etc.) Thus, it could be considered whether staging could be mentioned also in the title and/or in the last sentence of introduction section where something is missing in any case – typo?).
Author Response
I would like to thank the reviewer for their insightful comments and for the improvements they have already allowed us to make to the manuscript, on issues that we had overlooked at first glance.
With this new version of the paper attached, we wanted to address the shortcomings that were reported to us.
The idea of tracking which scanner acquired specific images is a truly interesting observation. At present, we have been unable to identify a portion of the accession number that could be easily associated with the scanner used, as in our institute it is generated when the patient is admitted, while assignment to the specific machine only occurs subsequently. For future studies, perhaps with a larger sample size, this represents an important point to work on.
Thank you very much for your comments on the quality of the article. We welcome any edits you may suggest.
We have addressed them the following way:
- Comment 1. We missed the appropriate reference, it has now been added;
- Comment 2. It was a mistype, we have fixed it;
- Comment 3. Indeed, the lowest SUV considered positive between all reports was 1,9. We have now changed the relative sentences.
- Comment 4. Thank you, indeed "Assessing the Potential Usefulness of FDG LAFOV-PET for Oncological Staging: Evaluation of Lesion Number and Uptake" should be a better title. And that was indeed a typo we had missed.
We thank you again for all the essential contributions.

This manuscript is a resubmission of an earlier submission. The following is a list of the peer review reports and author responses from that submission.
Round 1
Reviewer 1 Report
Comments and Suggestions for Authors
Dear Authors,
This is an interesting paper that focuses on an open field of research. Some issues are however present:
- in the title it could be useful to add the fact that this research focuses only on FDG;
- "On the other hand, patients who had undergone PET/CT for restaging after therapy or for monitoring during therapy were excluded, as were patients who had undergone previous PET scans for oncological diseases. The rationale for not including this cohort of patients was that the primary diagnosis could not be ascertained and that the semi-quantitative values of SUVmax may have been influenced by prior therapies". I agree with these choice, however the possible impact of LAFOV scanners in restaging setting or for therapy monitoring should be discussed in the discussion section;
- information about acquisition protocols, injected doses and reconstruction protocols used to perform PET scans is missing;
- information about the number of patients and PET scans included in the study and the precise number of different neoplasms is missing. A table with patients information could be added;
- line 179: I think that a word is missing;
- some neoplasms that are known to be false negative at FDG imaging have been included (e.g. HCC). This choide should be explained;
- the meaning of the word "MUN" is not specified the first time it was used but only in the discussion;
- if possible, an analysis of the real clinical impact of the lesions underlined by PET/CT could help to strengthen your findings. For example, in a patient with metastatic disease to the lung, adding 2 or 3 different lesions with PET in the same lobe will probably not impact its therapeutic managament.
- the meaning of "TBR" has not been specified;
Author Response
Comment 1: In the title it could be useful to add the fact that this research focuses only on FDG.
Response 1: Dear editor, thank you for your comments. The full title of this study is "Assessing the Potential Usefulness of LAFOV-PETs: Evaluation of Lesion Number and Uptake in Oncologic FDG PET/CT Scans". Would you suggest us to rephrase it in such a way that the FDG part stands out more?
Comment 2: "On the other hand, patients who had undergone PET/CT for restaging after therapy or for monitoring during therapy were excluded, as were patients who had undergone previous PET scans for oncological diseases. The rationale for not including this cohort of patients was that the primary diagnosis could not be ascertained and that the semi-quantitative values of SUVmax may have been influenced by prior therapies". I agree with these choice, however the possible impact of LAFOV scanners in restaging setting or for therapy monitoring should be discussed in the discussion section.
Response 2: Indeed, we think further studies assessing LAFOV scanners role in restaging settings would achieve relevant results. I have added a few sentences highlighting this.
Comment 3: Information about acquisition protocols, injected doses and reconstruction protocols used to perform PET scans is missing.
Response 3: Thank you, I had completely missed that. I have added the necessary info.
Comment 4: Information about the number of patients and PET scans included in the study and the precise number of different neoplasms is missing. A table with patients information could be added.
Response 4: We have listed the specific pathologies and their numbers between lines 197 and 203. If you think a table would add value to this section, it will be our pleasure to add it.
Comment 5: Line 179: I think that a word is missing.
Response 5: I have rephrased line 179, hoping to have fixed this issue.
Comment 6: Some neoplasms that are known to be false negative at FDG imaging have been included (e.g. HCC). This choide should be explained.
Response 6: I have now addressed this issue in the discussion section.
Comment 7: The meaning of the word "MUN" is not specified the first time it was used but only in the discussion.
Response 7: I have fixed this on line 188.
Comment 8: If possible, an analysis of the real clinical impact of the lesions underlined by PET/CT could help to strengthen your findings. For example, in a patient with metastatic disease to the lung, adding 2 or 3 different lesions with PET in the same lobe will probably not impact its therapeutic managament.
Response 8: Thank you, I have expanded on this part in the introduction.
Comment 9: The meaning of "TBR" has not been specified.
Response 9: Thank you, I have fixed this on line 273.

Reviewer 2 Report
Comments and Suggestions for Authors
Dear colleagues,
This study offers valuable insights into the potential clinical impact of Long Axial Field of View (LAFOV) PET technology in oncological staging, emphasizing its ability to enhance diagnostic sensitivity and influence treatment decisions. The authors’ focus on identifying a specific patient subset—those with 1-2 lesions and SUVmax values between 2 and 5—is both innovative and clinically relevant. However, several critical points require clarification and further exploration to strengthen the study’s conclusions.
1.Retrospective Design Limitations:
The study is based on retrospective data from SAFOV-PET scans without direct comparison to LAFOV-PET scans in the same cohort. This approach makes it challenging to definitively attribute the potential diagnostic improvements to LAFOV-PET technology, as other factors, such as scanner type, protocol variations, or patient heterogeneity, may have influenced the results. Could the authors elaborate on how these limitations might affect the interpretation of their findings?
2. (Page 3, Lines 109–112):Insufficient Connection to Therapeutic Decision-Making
The stated aim of the analysis is to identify clinical settings where LAFOV-PET could significantly influence therapeutic decision-making. While the study highlights that approximately 4.5% of the cohort might benefit from increased diagnostic sensitivity, it falls short of explicitly demonstrating how this sensitivity translates into changes in treatment strategies.
Author Response
Comment 1: Retrospective Design Limitations: The study is based on retrospective data from SAFOV-PET scans without direct comparison to LAFOV-PET scans in the same cohort. This approach makes it challenging to definitively attribute the potential diagnostic improvements to LAFOV-PET technology, as other factors, such as scanner type, protocol variations, or patient heterogeneity, may have influenced the results. Could the authors elaborate on how these limitations might affect the interpretation of their findings?
Response 1: Thank you, editor. Indeed, this is a limit of our study that is definitely worth pointing out, so I've added some sentences about it.
Comment 2: (Page 3, Lines 109–112):Insufficient Connection to Therapeutic Decision-Making
The stated aim of the analysis is to identify clinical settings where LAFOV-PET could significantly influence therapeutic decision-making. While the study highlights that approximately 4.5% of the cohort might benefit from increased diagnostic sensitivity, it falls short of explicitly demonstrating how this sensitivity translates into changes in treatment strategies.
Response 2: Thank you. Indeed, the other editor highlighted this part as well, and I quoted NSCLC guidelines in the introduction. I have now made further examples about breast cancer and melanoma in the discussion.

Round 2
Reviewer 2 Report
Comments and Suggestions for Authors
In this interesting paper, the authors assessing the potential usefulness of LAFOV-PETs to enhance diagnostic sensitivity in oncological care.
The key purpose was to assess in what proportion of patients suffering from oncological pathologies the new LAFOV-PET technology, due to its superior diagnostic sensitivity, could have a significant impact on the management of the therapeutic course of such patients compared to conventional SAFOV-PET scanners.
However, based on the study methods and results presented, there is insufficient evidence to support the claim that LAFOV-PET offers superior diagnostic sensitivity. The manuscript does not provide a direct comparison of diagnostic sensitivity between the two technologies, which is crucial to substantiate the claim made in the abstract and throughout the text.
Author Response
Dear editor, thank you for your observations, we will look into them before resubmitting the manuscript.